# Comparative Effectiveness Study of Home-Based Interventions to Prevent CA-MRSA Infection Recurrence

**DOI:** 10.3390/antibiotics10091105

**Published:** 2021-09-13

**Authors:** Jonathan N. Tobin, Suzanne Hower, Brianna M. D’Orazio, María Pardos de la Gándara, Teresa H. Evering, Chamanara Khalida, Jessica Ramachandran, Leidy Johana González, Rhonda G. Kost, Kimberly S. Vasquez, Hermínia de Lencastre, Alexander Tomasz, Barry S. Coller, Roger Vaughan

**Affiliations:** 1Clinical Directors Network, Inc. (CDN), New York, NY 10018, USA; shower@med.miami.edu (S.H.); brianna.dorazio@gmail.com (B.M.D.); CKhalida@CDNetwork.org (C.K.); jessicarama212@gmail.com (J.R.); johagoca1@gmail.com (L.J.G.); 2Center for Clinical and Translational Science, The Rockefeller University, New York, NY 10065, USA; maria.pardos-de-la-gandara@pasteur.fr (M.P.d.l.G.); evering@med.cornell.edu (T.H.E.); rhonda.kost@rockefeller.edu (R.G.K.); kimberly.vasquez@yale.edu (K.S.V.); barry.coller@rockefeller.edu (B.S.C.); rvaughan@rockefeller.edu (R.V.); 3Institut Pasteur, 75015 Paris, France; 4Weill Cornell Medicine, New York, NY 10065, USA; 5Metropolitan Hospital Center, New York City Health + Hospitals, New York, NY 10029, USA; 6Laboratory of Microbiology and Infectious Diseases, The Rockefeller University, New York, NY 10065, USA; lencash@mail.rockefeller.edu (H.d.L.); Alexander.Tomasz@rockefeller.edu (A.T.); 7Instituto de Tecnologia Química e Biológica (ITQB/UNL), 2780-157 Oeiras, Portugal

**Keywords:** methicillin-resistant *Staphylococcus aureus* (MRSA), antibiotic-resistance, skin and soft tissue infection (SSTI), community-based participatory research (CBPR), practice-based research network (PBRN), randomized clinical trial (RCT)

## Abstract

Recurrent skin and soft tissue infections (SSTI) caused by Community-Associated Methicillin-Resistant *Staphylococcus aureus* (CA-MRSA) or Methicillin-Sensitive *Staphylococcus aureus* (CA-MSSA) present treatment challenges. This community-based trial examined the effectiveness of an evidence-based intervention (CDC Guidelines, topical decolonization, surface decontamination) to reduce SSTI recurrence, mitigate household contamination/transmission, and improve patient-reported outcomes. Participants (n = 186) were individuals with confirmed MRSA(+)/MSSA(+) SSTIs and their household members. During home visits; Community Health Workers/Promotoras provided hygiene instructions; a five-day supply of nasal mupirocin; chlorhexidine for body cleansing; and household disinfecting wipes (Experimental; EXP) or Usual Care Control (UC CON) pamphlets. Primary outcome was six-month SSTI recurrence from electronic health records (EHR). Home visits (months 0; 3) and telephone assessments (months 0; 1; 6) collected self-report data. Index patients and participating household members provided surveillance culture swabs. Secondary outcomes included household surface contamination; household member colonization and transmission; quality of life; and satisfaction with care. There were no significant differences in SSTI recurrence between EXP and UC in the intent-to-treat cohort (n = 186) or the enrolled cohort (n = 119). EXP participants showed reduced but non-significant colonization rates. EXP and UC did not differ in household member transmission, contaminated surfaces, or patient-reported outcomes. This intervention did not reduce clinician-reported MRSA/MSSA SSTI recurrence. Taken together with other recent studies that employed more intensive decolonization protocols, it is possible that a promotora-delivered intervention instructing treatment for a longer or repetitive duration may be effective and should be examined by future studies.

## 1. Introduction

Methicillin-Resistant *Staphylococcus aureus* (MRSA) causes multi-drug resistant infections that pose serious clinical and public health challenges. Skin and soft tissue infections (SSTIs) [1,2] caused by MRSA carry significant morbidity and mortality, and impact patients, families, caregivers, and health-care institutions [3,4]. While studies comparing protocols for reducing healthcare-associated MRSA (HA-MRSA) infections [5] exist, those adapted for community-associated MRSA (CA-MRSA) SSTIs have provided mixed results [6,7,8,9,10,11]. CA-MRSA SSTIs commonly affect healthy, young individuals without exposure to healthcare risk factors or contacts [12]. (For a list of abbreviations used in this paper see Appendix A.)

Most CA-MRSA SSTIs are treated successfully in ambulatory care. However, treatment failure may result in risk of exposure and transmission to household and community members [9,13,14]. Even when primary treatment is successful, recurrent infections are common, ranging from 16% [13,15,16] to 43% [8,17]. Little research has examined the feasibility and effectiveness of implementing evidence-based interventions in primary care settings [18]. This trial tested two community-based interventions: (1) Usual Care (UC): CDC-Guidelines directed care [incision and drainage (I&D) and antibiogram-selected oral antibiotics] versus (2) Experimental (EXP): UC combined with universal household decolonization and environmental decontamination interventions based on the REDUCE MRSA Trial [5,19,20], provided in the home by Community Health Workers (CHW)/Promotoras. We evaluated the comparative effectiveness on SSTI recurrence rates (primary outcome) and household contamination, household member colonization and transmission, and patient-centered measures (pain, depression, quality of life, and care satisfaction; secondary outcomes) using a two-arm 1:1 randomized controlled trial (RCT). We hypothesized that participants assigned to EXP would experience fewer SSTI recurrences compared to UC.

## 2. Results

### 2.1. Baseline Comparisons: Experimental versus Usual Care

A total of 602 patients were assessed for eligibility: (Figure 1) 115 from three Federally Qualified Health Centers (FQHCs) and 487 from three Emergency Departments (EDs), of whom 421 (86.4%) provided informed consent, and 186 (44.2%) tested positive for MRSA or MSSA, and were invited to complete the baseline home visit (i.e., “intent-to-treat cohort”). Of 186 eligible consented participants, 120 (65%), completed baseline home visits.

Patients who did (n = 120) and did not (n = 66) complete the baseline home visit were similar (Appendix A), with equal proportions undergoing I&D (60.7%). Both groups exhibited similar dermatologic symptoms, with no differences in lesion location, size, type, purulence, or signs/symptoms of SSTIs (Appendix A). We also compared those who completed the allocated intervention and had 6-month follow-up data (n = 92) to those who received the allocated intervention but did not complete the 6-month follow-up survey (n = 28, Appendix A). Rates of I&D, MRSA/MSSA, and recruitment source (FQHC vs. ED) were similar.

No statistically significant differences were detected between EXP and UC in baseline demographics, comorbidity, baseline occupational or environmental or social exposures (Table 1 or in the proportion of household members who participated in the study About half (52.5%) spoke English as their primary language and 36.4% spoke Spanish as their primary language. Microbiological and dermatologic characteristics, and health care utilization were similar (Appendix A), but MRSA+ wounds were more common among patients randomized to EXP (66.1%) as compared to UC (32.1%, *p* = 0.0004).

### 2.2. SSTI Recurrence

We used logistic regression analyses to detect treatment group differences on EHR-documented SSTI recurrence at six-month follow-up (Table 2). We conducted initial main effects analysis using all randomized subjects (n = 186, “intent-to-treat cohort”). Based on the sensitivity analysis, we conducted all further analyses using completed cases (n = 119, “analysis cohort”). The ITT analysis demonstrated no significant differences in infection recurrence between EXP (11.1%) and UC (10.7%; OR = 1.4, 95% CI = 0.51–3.5). Likewise, when examining the analysis cohort, there were no statistically significant differences between EXP and UC on EHR-documented SSTI recurrence: 11.1% of EXP vs. 10.7% of UC had a documented SSTI recurrence (OR = 1.14, 95% CI = 0.35, 3.6; Table 2).

Differences in SSTI recurrence based on self-report (Table 2) showed a trend in the opposite direction: EXP reported a greater recurrence rate at 6 months (22.2%) than UC (7.5%, OR = 3.5, 95% CI = 0.89, 13.8). Given the differences between EHR-documented and self-reported recurrence we examined concordance by treatment group and by baseline wound characterization. 15.4% of participants with a self-reported SSTI recurrence also had a documented clinical SSTI recurrence. Retrospective self-report of pre-study infections found 30.5% of participants reported ≥1 prior SSTI, whereas 90.7% of participants had documented pre-study SSTIs in their EHRs, indicating poor concordance between EHR and self-report.

The ITT analysis resulted in similar trends when restricted to the enrolled analysis cohort. We also added one additional unplanned analysis comparing the EHR-measured 6-month SSTI outcome with an “observation only control group” (n = 66) comprised of eligible individuals with a confirmed MRSA+/MSSA+ wound culture who had initially consented to participate, but did not complete the baseline home visit, so their randomization assignment was never disclosed to these participants or study staff (Appendix A). These participants did not receive any intervention or assessment beyond what was extracted from the EHR, and had no further interactions with research staff, although they continued to receive care where they were recruited. The observed prospective SSTI recurrence rate at 6-month EHR review for the “observation only control group” (10.5%) was similar to those of the EXP (11.1%) and UC (10.7%) groups.

### 2.3. Index Patient Colonization

Index patient colonization (nares, axilla, and groin) was measured at baseline and three months. Similar baseline colonization rates were observed: nares (EXP = 41.3% vs. UC = 32.1%), axilla (EXP = 34.9% vs. UC = 32.1%), groin (EXP = 49.2% vs. UC = 44.6%; Figure 2a). *S. aureus* was recovered at baseline from at least one site (Figure 2b) in most patients (EXP = 74.6% vs. UC = 66.1%). While most patients were colonized at one site (EXP = 34.9% vs. UC = 35.7%), some were colonized at two (EXP = 28.6% vs. UC = 17.9%) or three sites (EXP = 11.1% vs. UC = 12.5%).

Three months post-intervention, there was an overall reduction in colonization, with *S. aureus* less frequently recovered from all body sites. Implementation of the decolonization intervention demonstrated that colonization rates in EXP were non-significantly reduced at three months for nares (OR = 0.41, 95% CI = 0.16–1.04), axilla (OR = 0.77, 95% CI = 0.31–1.91) and groin (OR = 0.53, 95% CI = 0.24–1.20), whereas there were little decreases in UC (Figure 2a). Overall, the increase in participants with no organisms detected at three months was greater in EXP (36.5% point increase) as compared to UC (17.2% point increase; Figure 2b). We did not observe any increase in mupirocin resistance between baseline (3%) and three months (0%; data not shown).

### 2.4. Patient-Centered Outcomes

No intervention-related behavioral changes were observed, including infection prevention knowledge and hygiene, prevention self-efficacy, decision-making autonomy, mupirocin and chlorhexidine adherence, quality of life (QoL) or patient satisfaction (Appendix A).

### 2.5. Household Surfaces Contamination

At baseline, the most frequently contaminated surfaces included: toilet seat (EXP = 52.4% vs. UC = 53.6%), bedroom floor (EXP = 49.2% vs. UC = 60.7%), and kitchen floor (EXP = 60.3% vs. UC = 58.9%; Figure 3a), and most households had ≥1 contaminated surfaces (EXP = 96.8% vs. UC = 96.4%). Both groups showed similar reductions in environmental contamination (Figure 3b). There were no differences in reductions in proportions of households with ≥1 contaminated surfaces, EXP (96.8% to 60.3%) vs. UC (96.4% to 66.1%). Linear regression examining the average difference in numbers of contaminated surfaces (0–13) showed that there were 0.31 fewer contaminated surfaces at follow-up in EXP versus UC, after adjusting for baseline number of contaminated surfaces (*p* = 0.08; data not shown). Multivariate models with treatment allocation and number of household surfaces did not reveal any associations among these environmental-level factors and SSTI recurrence.

### 2.6. Household Members Colonization and Reported Infection

Consenting household members were screened for colonization (nares, axilla, groin) at baseline and three months. 64.1% of household members provided surveillance cultures (EXP = 67.5%, UC = 60.2%, *p* = 0.34; see Figure 4a, b). Among household members, similar reductions in proportions of colonized sites were seen for nares (EXP 27.0% to 17.5% vs. UC 23.2% to 17.9%), axilla (EXP 17.5% to 9.5% vs. UC 14.3% to 10.7%) and groin (EXP 28.6% to 19.1% vs. UC 21.4% to 19.6%; Figure 2a). There was a non-significant reduction in household member colonization for EXP (9.5% to 3.2%) versus UC (8.9% to 5.4%). MSSA+ household members were similar at baseline and three-month follow-up in both EXP and UC, with no observed reductions in percentages of household members colonized by MSSA (Figure 2b).

### 2.7. Subgroup Considerations/Heterogeneity of Treatment Effects (HTE)

We conducted pre-specified subgroup analyses using logistic regression with treatment assignment and with each subgroup coded as dummy variables. Consistent with previous studies [21], foreign-born participants were more likely to have MSSA+ than MRSA+ wound cultures. Other HTE subgroup comparisons (wound culture type (MRSA vs. MSSA), birthplace (USA vs. non-USA), household contamination levels (low vs. high), household members colonization (present vs. absent), pets living in the household (present vs. absent), recruitment site (ED vs. FQHC), and baseline I&D treatment (yes vs. no)) revealed no statistically significant differences for EHR-documented SSTI recurrence (data not shown).

## 3. Discussion

This study adapted and implemented an effective hospital ICU-based intervention [5] into the community. We examined the comparative effectiveness of usual care: CDC/Guidelines-directed care [19,20] versus an experimental intervention: UC combined with universal decolonization and environmental decontamination [5,19,20]. Results suggest that this was a null trial. There were few observed hypothesized intervention-related differences for clinical, microbiological and patient-centered outcomes.

The primary and secondary outcomes analyses indicated that EXP fared as well as UC. Despite a significant baseline difference in MRSA-positive (more common in EXP) and MSSA-positive (more common in UC) cultures which we had not anticipated and therefore did not stratify during randomization, we saw no evidence that this baseline imbalance affected the analyses.

Interestingly, the overall study rate of SSTI recurrence was substantially lower (10.8%) than in previous studies [22,23], although a recent cohort study reported a comparable 3-month recurrence rate (10.3%) [9]. It is possible that high study rates of I&D plus oral antibiotics contributed to the lower-than-expected SSTI recurrence rate. One recruitment site, a large public hospital ED, predominated, which might explain a lower-than-expected recurrence since treatments used there have been demonstrated effective in preventing treatment failure and SSTI recurrence [9,24]; thus, suggesting a statistical floor effect hindered detection of differences in recurrence. Similarly, the low event rates of household and environmental outcomes reduced the study’s power to detect significant treatment effects. Finally, EXP were more likely to have no detectable *S. aureus* colonization at three months, but again the difference was not significant.

The environmental persistence of *S. aureus* and as a colonizer despite active eradication efforts is well-documented and multifactorial [8,25,26,27] and likely modulated by interactions inside the household and surrounding community [9,15,28,29]. Overall, the percent of households with no environmental contamination increased substantially (from 3.2% to 39.7%). However, 60.3% of households were still contaminated and enhanced antimicrobial measures were not more effective [8] than standard patient education and UC. This reflects the inherent challenges of eradicating this opportunistic pathogen in residential settings. While previous studies have demonstrated higher rates of colonization and contamination are associated with recurrence [17,29,30], there are conflicting reports of whether reduction in bioburden translates into less recurrence [18,31,32]. Previous interventions to reduce *S. aureus* carriage and SSTI recurrence provide mixed results [6,8,10,11,18,32,33,34]. Interestingly, Golding et al. showed community education focusing on patient and household hygiene decreased SSTI incidence [35]. The universally distributed *Living with MRSA* pamphlet [36] likely contributed to total study recurrence reduction, potentially obscuring the intervention impact.

### 3.1. Study Limitations

This study’s results likely reflect unmeasured and uncontrolled variables, including adherence and effectiveness of bioburden reduction and microbial dynamics in an open system [26]. Additionally, there were significantly more MRSA+ wounds in EXP vs. UC despite randomization. This higher MRSA bioburden may have made decolonization and decontamination more challenging [29], thereby obfuscating any study treatment-related differences. Since EXP participants were aware of the intervention methods, it is plausible that their sensitization led participants to focus on minor skin symptoms that were ignored by UC, resulting in higher self-reported SSTI recurrence.

In order to minimize the burden of multiple home visits and increase study participation, follow-up sampling took place at three months, rather than immediately following initial decolonization and decontamination. Therefore, the immediate effectiveness of the experimental protocol was not measured. It is possible that organisms detected after three-months were not eradicated at baseline since *S. aureus* can exhibit long-term survival on surfaces [37]. However, host recolonization and replenishment of the environment over time is also possible.

### 3.2. Future Directions

Implementing common hospital cleaning protocols [5,38] within a community setting was a formidable challenge and warrants further investigation. Future studies should stratify on infection wound culture type (MRSA(+) vs. MSSA(+)) prior to randomization, to ensure balance in treatment assignment, and may wish to power each subgroup sufficiently for SSTI recurrence, as well as measure effects of active antimicrobial measures immediately following completion of the regimen. While nasal decolonization routine herein is standard practice [20], it is possible that a longer and/or more potent decolonization protocol is required to reduce MRSA recurrence [11,23,30,39]. In fact, recent studies implementing a similar yet more intense decolonization intervention decreased MRSA infection recurrence [11,39]. These data, combined with our data, indicate that greater frequency and longer duration of decolonization may be required.

## 4. Materials and Methods

This RCT included practicing clinicians, patients, clinical and laboratory researchers, NY-based Federally Qualified Health Centers (FQHCs) and community hospital Emergency Departments (EDs). The stakeholder research collaborative expands an earlier partnership, the Community-Acquired MRSA Project (CAMP1) which developed research to address CA-MRSA [21,40,41,42]. The study was approved by Institutional Review Boards at Clinical Directors Network and Rockefeller University.

### 4.1. Study Setting

Three NYC FQHCs and three EDs recruited participants presenting with an SSTI with culture-positive MRSA or MSSA.

### 4.2. Participants

**Inclusion/Exclusion Criteria**: Participants included were: (1) between 7–70 years, (2) fluent in English or Spanish, (3) planning to receive follow-up care at the FQHC or ED, (4) presenting with SSTI signs/symptoms (5), had laboratory-confirmed baseline wound culture positive for MRSA or MSSA (also a significant cause of SSTIs; (6) willing/able to provide informed consent, and (7) willing to participate in two home visits. Participants were excluded if they were: (1) unwilling to provide informed consent, (2) acutely ill or visibly distressed (for example, crying, wheezing, bleeding, screaming or shaken), and/or (3) unable to participate in home visits or in a discussion about the study.

### 4.3. Study Protocol

**Screening:** Patients seeking care for an SSTI were identified via the site’s electronic health record (EHR) or clinical dashboard. CHWs flagged patients for treating clinicians to determine eligibility.

**Procedures**: Recruitment, informed consent, and baseline clinical assessments were conducted by trained CHW/Promotoras with clinicians and FQHC/ED staff. Following written informed consent, clinicians conducted baseline clinical assessments and collected wound cultures. Wound and surveillance (nasal, axilla, and groin) cultures were sent to one commercial clinical laboratory (BioReference) for culture, antibiotic susceptibility testing, and speciation. Additional molecular epidemiologic testing was carried out by Rockefeller’s Laboratory of Microbiology and Infectious Diseases [41] Mupirocin susceptibility was tested using E-test strips (bioMérieux, Durham, NC) following CLSI recommendations [43].

All participants received clinician-directed standard-of-care treatment, including I&D and/or oral antibiotics. If I&D was performed, a sample of purulent drainage material was obtained; if I&D was not indicated, the clinicians took a swab of the wound or, if the wound was weeping or draining, obtained purulent material. During the same visit, on-site CHW/Promotoras scheduled the home visit. If a CHW/Promotora or other research staff member was not present, per “warm hand off” procedures [44], clinical staff informed participants that a CHW/Promotora would telephone them.

When culture lab results became available (2-3 days later), medical staff disclosed MRSA/MSSA status. CHW/Promotoras then called each participant to inform them of eligibility. For those with MRSA+ or MSSA+ SSTIs, CHW/Promotoras confirmed the baseline telephone interview appointment and baseline home visit.

### 4.4. Assessments

**Baseline Assessments:** Appendix A details the full assessment protocol. A Baseline Telephone Questionnaire including demographics, medical history, comorbidity, social, occupational and environmental exposures, household composition and patient-centered outcomes (Appendix A) was administered in English or Spanish.

The Baseline Home Visit Assessment (T1) captured: (1) consenting household members’ demographics, comorbidities, SSTI history, and index patient and household member personal hygiene, (2) household sharing behaviors, (3) collection and retrieval of self-sampled surveillance cultures from the index patient and consenting household member, (4) census of the numbers of rooms, household inhabitants and regular visitors, (5) 13 samples obtained from high touch/high traffic household environmental surfaces (see Figure 4a) using ESwabs™ (Copan Diagnostics, Inc., Murrieta, CA, USA).

CHWs/Promotoras oversaw participants’ and household members’ self-sampling of nares, inguinal folds, and axillae. Prior to randomization, participants received the educational pamphlet, “Living with MRSA”, available in English and Spanish [36].

**Follow Up Assessments:** Home visits occurred at three-months (T3). Interim telephone assessments at one-month (T2) and six-month follow-up (T4) were conducted (Appendix A). Reviews of EHRs were conducted at T4 to record SSTI recurrence (i.e., one or more discrete clinical SSTI(s) at the same or new site in addition to the baseline infection during the six months following trial enrollment). Patient-reported SSTI recurrence was also recorded. While we had planned to combine SSTI recurrence data from EHR and self-reports, participants’ self-reports had poor concordance with EHR-based clinician reports, so we limited the main analysis to clinician-documented SSTI recurrence. See Appendix A.

**Randomization:** After baseline questionnaires and household sampling data were collected, and while still at the home, CHWs/Promotoras opened a sealed opaque envelope containing the computer-generated randomization (1:1), overseen by the offsite, blinded study statistician (JCdR). Participants were not stratified during randomization based on recruitment site or pathogen (MRSA or MSSA). If randomized to UC, the CHW/Promotoras explained the timeline for remaining assessments and concluded the visit. No further intervention materials were provided to UC.

**Interventions:** Modeled on the REDUCE MRSA trial [5], EXP received CDC and Infectious Disease Society of America guidelines-directed usual care [19,38] combined with universal household decolonization and environmental decontamination educational interventions [5] and materials from CHWs/Promotoras. They received detailed verbal, written, and demonstrated instructions of the five-day protocol of twice-daily application of mupirocin ointment to the anterior nares with a clean cotton applicator [45], once-daily Hibiclens^®^ (chlorhexidine gluconate solution 4% w/v) whole body wash [8,11,19,46], and household decontamination instruction including: (1) proper handwashing technique, (2) laundering bed linens and pillows in warm water every other day, and (3) disinfection of “high touch” environmental surfaces with disposable disinfecting wipes [38,47].

**Retention and Withdrawal:** Participants who withdrew were asked to provide reason(s) for withdrawal (see CONSORT Diagram, Figure 1 above). We attempted to reach all participants until the trial completion date.

### 4.5. Sample Size Calculations and Power

Sample size estimation was based on SSTI recurrence rate from CAMP1, where 33.3% experienced a recurrence during the six months following their index SSTI [21]; previous studies of HA-MRSA reported reductions between 30% and 55% [48,49,50]. To achieve 80% power at 5% significance level in a two-sided Chi Square/Fisher’s Exact test for recurrence at 6-month follow-up, estimated sample size requirement was 120 participants (60 per group).

### 4.6. Analytical and Statistical Approaches

Chi-square tests were applied for comparison of proportions and *t*-tests were used for continuous quantitative variables. All primary main effects outcomes analyses (n = 186) were evaluated according to “Intent-to-Treat” (ITT). Subsequent planned analyses included patients who completed the baseline home visit (n = 119, “analysis cohort”) and pre-specified subgroups. Logistic regression and generalized linear mixed effect models were used for hypothesis testing. All analyses were conducted with SAS (Version 9.3) or R (Version 3.0). Revised sample size calculations based on a conditional power analysis using the lower observed event rate indicate that the study still had a power > 0.80 to test a 50% reduction given the baseline recurrence rate of 11% (rather than 33%).

### 4.7. Missing Data/Sensitivity Analyses

Although we observed a small proportion of missing data, data were assumed to be “missing not at random” (MNAR) so we used a sensitivity approach rather than multiple imputation. No meaningful or statistically significant differences were revealed between the original versus sensitivity analysis results.

## 5. Conclusions

This trial aimed to understand systems-, patient-, pathogen-, and environmental-level factors associated with SSTI recurrence and household transmission (Appendix A). Home visits presented a major challenge. Although the perceived (or actual) intrusiveness of home visits proved difficult to overcome, the “warm hand-off” strategy facilitated a modest improvement in home visit completion rates. The lower-than-expected six-month recurrence rate (10.9% here as compared to our previous observational study of 33.3% [21]) may have limited the power to identify a treatment effect. Multiple well-designed studies conducted in different settings have all converged on similar findings: decolonization and decontamination can be accomplished in the household to varying degrees, but extensive, long-term decontamination may be required to achieve a medically meaningful reduction in disease recurrence. These findings suggest that other mechanisms may also contribute to the disconnect between exposure and outcome, such as intrinsic host factors including immunologic competence, as well as perturbations of the host and environmental microbiomes. These factors warrant further observational and experimental studies.

## Figures and Tables

**Figure 1 antibiotics-10-01105-f001:**
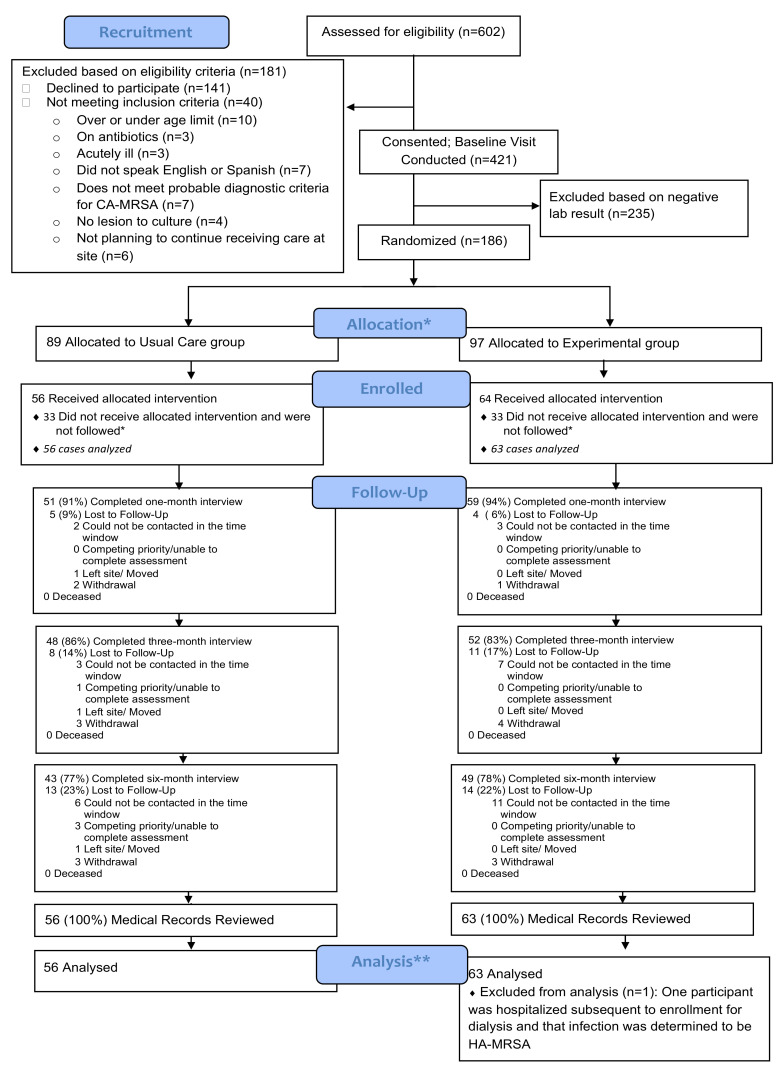
CONSORT Diagram. * “Intent-to-treat” (ITT) cohort (n = 186) ** “Analysis” cohort (n = 119). An additional n = 66 participants were eligible for participation based on having either MRSA+/MSSA+ wound culture and provided informed consent; however, these participants did not complete the baseline home visit, and therefore never received the intervention. Since these participants had consented to be followed, their six-month chart review data were extracted and are provided as an additional “observation-only control group” (n = 63 of these participants were analyzed, for a 95.5% response rate). Upon subsequent chart review, one patient was determined subsequently to have met criteria for HA-MRSA and considered ineligible and these data were removed from the analysis, leaving n = 119 (“analysis cohort”). Data collection spanned from 11/01/2015 to 11/25/2017.

**Figure 2 antibiotics-10-01105-f002:**
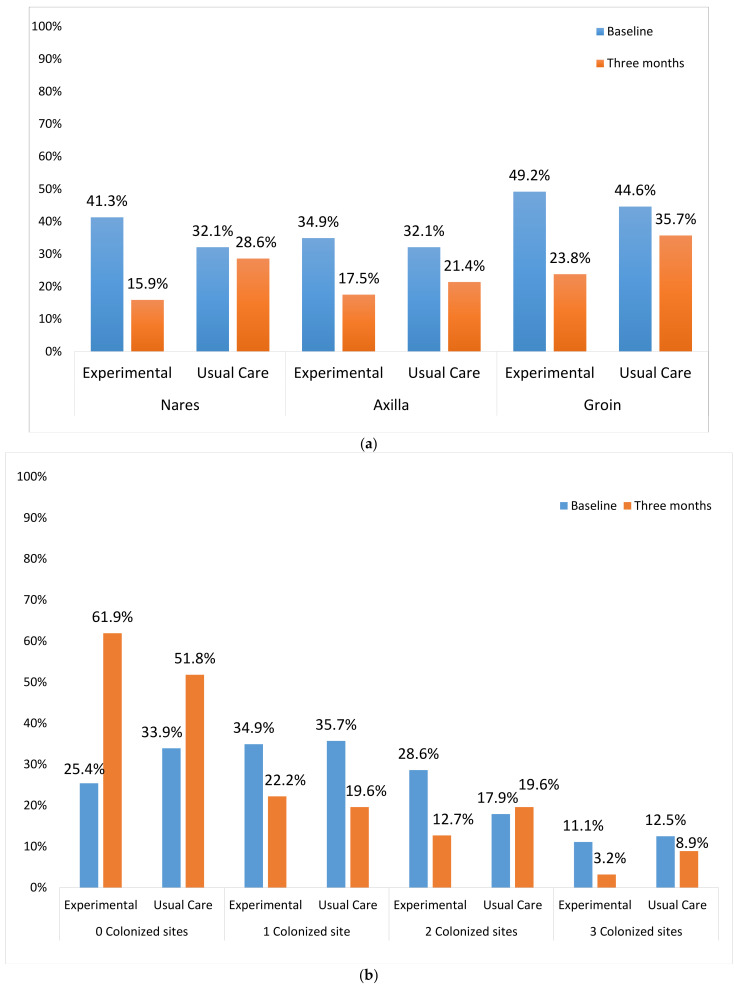
(**a**) Proportion of Index Patient *S. aureus* Colonization by Colonization Site at Baseline and Three Month Follow-up Household Visits. (**b**) Proportion of Index Patient S. *aureus* Colonization by Number of Colonized Sites at Baseline and Three Month Follow-up Household Visits.

**Figure 3 antibiotics-10-01105-f003:**
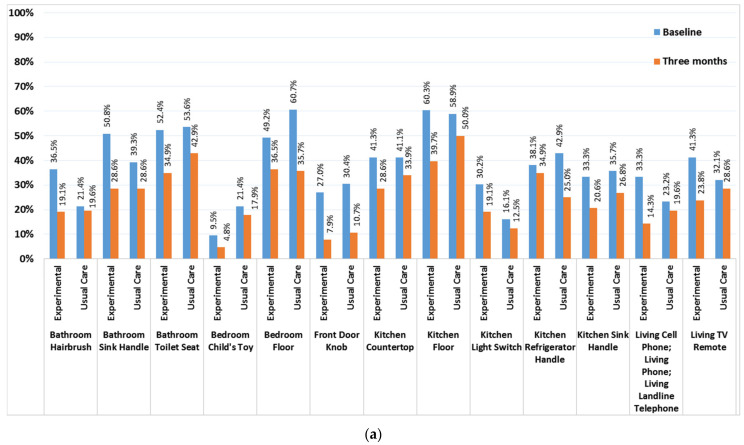
(**a**) Specific Household Surfaces Contaminated by *S. aureus* by Treatment Group at Baseline and Three Months Household Visit. (**b**). Numbers Household Surfaces Contaminated by *S. aureus* by Treatment Group at Baseline and Three months.

**Figure 4 antibiotics-10-01105-f004:**
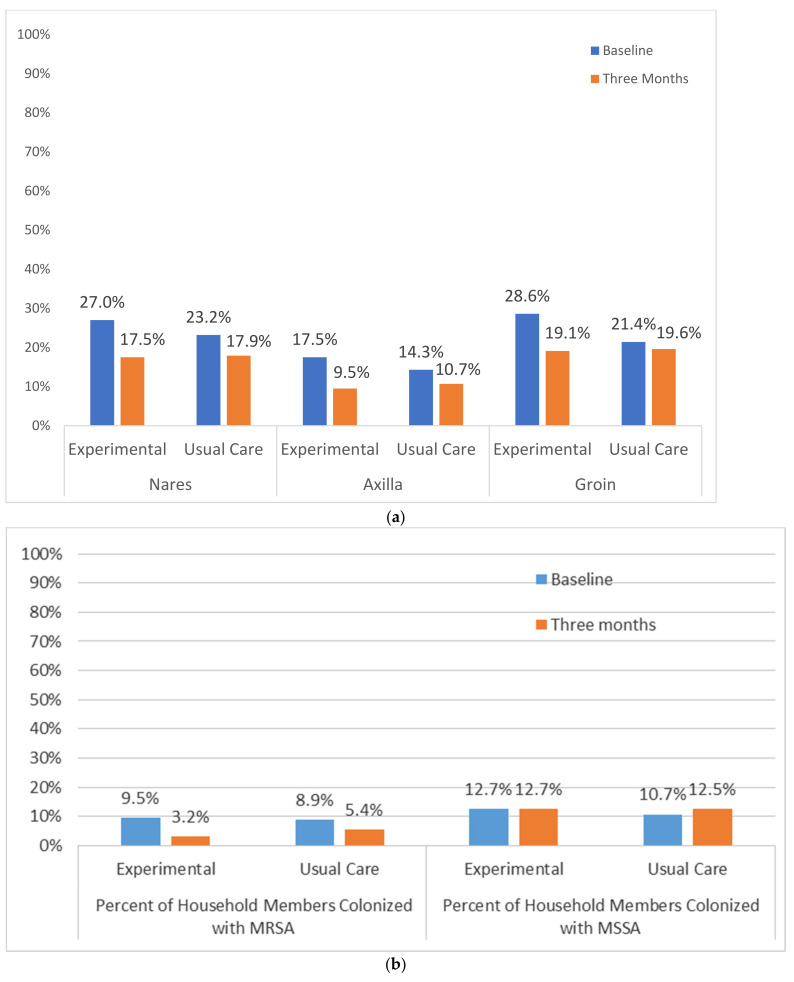
(**a**) Proportion of household members colonized with *S. aureus*, by colonization site at baseline and three month follow-up household visits. (**b**) Household members colonized with MRSA vs. MSSA by treatment group at baseline and three-month follow-up household visits. Note: In addition to the Index Patient information provided above, 64% of household members participated in the study and provided surveillance cultures (EXP = 67.5%, UC = 60.2%, *p* = 0.34) There were no significant differences by treatment group in the number of co-residents in households where the index patient had a MRSA wound (EXP = 2.4 vs. UC = 3.4, *p* = 0.06).

**Table 1 antibiotics-10-01105-t001:** Comparison of Study Participants at Baseline for (a) demographics, (b) clinical comorbidity and health care utilization, and (c) occupational and social exposures.

(a) Demographic Characteristics by Treatment Group at Baseline			
	Total	Experimental	Usual Care
	(n = 119)	(n = 63)	(n = 56)
**AGE, n (%)**			
7–18 years	12 (10.1)	5 (7.9)	7 (12.5)
19–64 years	103 (86.6)	55 (87.3)	48 (85.7)
over 65	4 (3.4)	3 (4.8)	1 (1.8)
Mean (SD)	38.1 (14.9)	39.5 (15.4)	36.5 (14.4)
**GENDER, n (%)**			
Female	47 (39.5)	22 (34.9)	25 (44.6)
Male	72 (60.5)	41 (65.1)	31 (55.4)
**ETHNICITY, n (%)**			
Hispanic or Latino	72 (64.9)	37 (62.7)	35 (67.3)
Not Hispanic or Latino	36 (32.4)	21 (35.6)	15 (28.9)
Prefer not to answer	3 (2.7)	1 (1.7)	2 (3.9)
**RACE, n (%)**			
American Indian or Alaska Native	1 (1.0)	0 (0)	1 (1.8)
Asian	0 (0)	0 (0)	0 (0)
Black or African American	27 (22.7)	14 (22.2)	13 (23.3)
White	22 (18.5)	13 (20.6)	9 (16.1)
More than one race	21 (17.6)	11 (17.5)	10 (17.9)
Prefer not to answer/Unknown	48 (40.3)	25 (39.7)	23 (41.1)
**BIRTHPLACE, n (%)**			
One of the 50 U.S. States	70 (58.5)	40 (63.5)	54 (53.6)
Puerto Rico	7 (5.9)	5 (7.9)	2 (3.6)
“Other” Country	42 (35.3)	18 (28.6)	24 (42.9)
**Birthplace, if not USA, per self-report, n (%)**			
Africa (Ivory Coast, Senegal, Other Unspecified)	3 (7.1)	2 (11.1)	1 (4.2)
South America (Colombia, Ecuador)	5 (11.9)	4 (22.2)	2 (8.3)
North/Central America (Dominican Republic, Guatemala, Honduras, Mexico, Panama)	29 (69.0)	12 (66.7)	17 (70.8)
Europe (Russia, Ukraine)	2 (4.8)	0 (0)	2 (8.3)
Asia (Uzbekistan, Yemen)	3 (7.1)	1 (5.6)	2 (8.3)
**Length of time in the USA (if non-USA born), n (%)**			
Less than 10 years	96 (80.7)	52 (82.5)	44 (78.6)
10 years and over	23 (19.3)	11 (17.5)	12 (21.4)
Years in US, Mean (SD)	17.2 (13.7)	18 (14.4)	16.5 (13.5)
**LANGUAGE SPOKEN AT HOME, n (%)**			
English	62 (52.5)	32 (51.6)	30 (53.6)
Spanish	43 (36.4)	24 (38.7)	7 (12.5)
Other (Portuguese, Ghanaian)	13 (11.0)	6 (9.7)	19 (33.9)
**MARITAL STATUS, n (%)**			
Married/Living with partner	44 (37.3)	22 (34.9)	22 (40.0)
Widowed	5 (4.2)	4 (6.4)	1 (1.8)
Divorced	6 (5.1)	3 (4.8)	3 (5.5)
Separated	7 (5.9)	4 (6.4)	3 (5.5)
Never Married	56 (47.5)	30 (47.6)	26 (47.3)
Not reported	1	0	1
**HIGHEST DEGREE OR LEVEL OF SCHOOL COMPLETED, n (%)**			
Below high school	46 (39.0)	24 (38.7)	22 (39.3)
High school or over	72 (61.0)	38 (61.3)	34 (60.7)
Not reported	1	1	0
**COMBINED FAMILY INCOME, n (%)**			
Less than $10,000	33 (27.7)	18 (28.6)	15 (26.8)
$10,000 or greater	57 (47.9)	31 (49.2)	26 (46.4)
Not reported	29 (24.4)	14 (22.2)	15 (26.8)
**TYPE OF INSURANCE, n (%)**			
Private Insurance	10 (8.5)	5 (7.9)	5 (9.1)
Medicare	11 (9.4)	7 (11.1)	4 (7.3)
Medicaid	56 (47.5)	35 (55.6)	21 (38.2)
HMO	1 (0.9)	1 (1.6)	0 (0)
Military or Veteran	1 (0.9)	0 (0)	1 (1.8)
None	26 (22.0)	11 (17.5)	15 (27.3)
Other	13 (11.0)	4 (6.4)	9 (16.4)
**EMPLOYMENT HISTORY, n (%)**			
Currently Employed	56 (47.5)	29 (46.8)	27 (48.2)
**EXTENT OF EMPLOYMENT, n (%)**			
Full time	33 (58.9)	17 (58.6)	16 (59.3)
Part time	23 (41.1)	12 (41.4)	11 (40.7)
(**b**) **Comorbidity and Health Care Utilization** **by Treatment Group at Baseline**			
**BODY MASS INDEX (BMI), n (%)**			
Underweight (≤18.5)	3 (2.8)	1 (1.8)	2 (3.9)
Normal weight (18.6–24.9)	32 (29.6)	16 (28.1)	16 (31.4)
Overweight (25–29.9)	34 (31.5)	17 (29.8)	17 (33.3)
Obese (≤30)	39 (36.1)	23 (40.4)	16 (31.4)
Mean (SD)	28.8 (6.5)	29.2 (6.3)	28.3 (6.8)
**CO-MORBIDITY, n (%)**			
Abscess/Boil	108 (90.8)	57 (90.5)	51 (91.1)
Alcohol Abuse	4 (3.4)	3 (4.8)	1 (1.8)
Arteriosclerotic Cardiovascular Disease or Coronary Artery Disease	8 (6.7)	3 (4.8)	5 (8.9)
Asthma	20 (17.1)	12 (19.4)	8 (14.6)
Chronic Liver Disease	3 (2.5)	2 (3.2)	1 (1.8)
Chronic Renal Insufficiency	6 (5.0)	4 (6.4)	2 (3.6)
Chronic Skin Breakdown	6 (5.1)	3 (4.8)	3 (5.4)
Current Smoker	33 (28.2)	17 (27.4)	16 (29.1)
Cerebrovascular Accident (CVA)/Stroke (Not Transient Ischemic Attack (TIA)	6 (5.1)	3 (4.8)	3 (5.5)
Cystic Fibrosis	0 (0)	0 (0)	0 (0)
Decubitus/Pressure Ulcer	2 (1.7)	2 (3.2)	0 (0)
Dementia	1 (0.9)	0 (0)	1 (1.8)
Diabetes	29 (24.6)	16 (25.8)	13 (23.2)
Emphysema/Chronic obstructive Pulmonary Disease (COPD)	3 (2.7)	3 (5.1)	0 (0)
Heart Failure	4 (3.4)	4 (6.4)	0 (0)
Hematologic Malignancy	1 (0.8)	1 (1.6)	0 (0)
Hemiplegia/Paraplegia	1 (0.8)	0 (0)	1 (1.8)
HIV or AIDS	4 (3.4)	1 (1.6)	3 (5.4)
Immunosuppressive Therapy	3 (2.5)	2 (3.2)	1 (1.8)
Intravenous Drug Use	5 (4.2)	4 (6.4)	1 (1.8)
Metastatic Solid Tumor	2 (1.7)	2 (3.2)	0 (0)
Obesity	16 (13.7)	8 (13.1)	8 (14.3)
Other Drug Use	9 (7.6)	4 (6.5)	5 (8.9)
Peptic Ulcer Disease	4 (3.4)	3 (4.8)	1 (1.8)
Peripheral Vascular Disease	1 (0.8)	1 (1.6)	0 (0)
Rheumatoid Arthritis	1 (0.9)	0 (0)	1 (1.9)
Sickle Cell Anemia	2 (1.7)	2 (3.2)	0 (0)
Systemic Lupus Erythematosus	2 (1.7)	1 (1.6)	1 (1.8)
**HEALTHCARE UTILIZATION: Six Months Prior to Baseline, n (%)**			
Hospitalized	37 (31.1)	24 (38.1)	13 (23.2)
Nights hospitalized, Mean (SD)	3.0 (8.3)	3.1 (6.8)	2.9 (9.8)
Visited ED/Urgent care facility	106 (89.1)	54 (85.7)	52 (92.9)
Visits to ED/Urgent care facility, Mean (SD)	1.9 (2.2)	2.0 (2.6)	1.9 (1.5)
**Doctor Visits, n (%)**			
≤3	88 (74.6)	47 (75.8)	41 (73.2)
4 to 8	18 (15.3)	8 (12.9)	10 (17.9)
≥9	12 (10.2)	7 (11.3)	5 (8.9)
Mean (SD)	3.1 (5.3)	3.1 (5.6)	3.1 (5.1)
**MEDICAL HISTORY, n (%)**			
Prior Treatment for Same Lesion	32 (27.1)	14 (22.6)	18 (32.1)
Family/Friends with Same Lesion	15 (12.6)	8 (12.7)	7 (12.5)
Had Lesion While in School	13 (10.9)	5 (7.9)	8 (14.3)
Had Lesion While Working	23 (19.3)	12 (19.1)	11 (19.6)
(**c**) **Occupational and Social Exposures** **by Treatment Group at Baseline**			
**OCCUPATIONAL EXPOSURE, n (%)**			
Healthcare Employee	0 (0)	0 (0)	0 (0)
Nursing Home Employee	1 (0.8)	1 (1.6)	0 (0)
Daycare Center Employee	2 (1.7)	1 (1.6)	1 (1.8)
Correctional Facility Employee	1 (0.8)	1 (1.6)	0 (0)
Animal Facility Employee	1 (0.8)	1 (1.6)	0 (0)
ENVIRONMENTAL EXPOSURES, n (%)			
Handwashes per Day:			
0–1 time	0 (0)	0 (0)	0 (0)
2–3 times	11 (9.2)	6 (9.5)	5 (8.9)
4–6 times	44 (37.0)	23 (36.5)	21 (37.5)
7–10 times	25 (21.0)	12 (19.1)	13 (23.2)
>10 times	39 (32.8)	22 (34.9)	17 (30.4)
In Month Prior to Baseline:			
Surgery	8 (6.7)	6 (9.5)	2 (3.6)
Wounds, cuts, abrasions	33 (27.7)	16 (25.4)	17 (30.4)
Spent time in hospital	39 (32.8)	24 (38.1)	15 (26.8)
International Travel	7 (5.9)	4 (6.4)	3 (5.4)
Incarceration	0 (0)	0 (0)	0 (0)
Lived in Dormitory	3 (2.5)	0 (0)	3 (5.4)
Lived in Military Barracks	0 (0)	0 (0)	0 (0)
Taken Antibiotics	105 (88.2)	55 (87.3)	50 (89.3)
Spent time at Daycare Center	6 (5.1)	4 (6.5)	2 (3.6)
Played Contact Sports	17 (14.3)	7 (11.1)	10 (17.9)
Had Pets in the House	40 (33.6)	19 (30.2)	21 (37.5)
SOCIAL NETWORK, n (%)			
How many people live in your house?			
Lives alone (one person household)	7 (5.9)	5 (7.9)	2 (3.6)
2–3 people	52 (43.7)	31 (49.2)	21 (37.5)
4 or over	60 (50.4)	27 (42.9)	33 (58.9)
Sharing Characteristics			
Shared Bedroom or Sleeping Space	62 (52.1)	31 (49.2)	31 (55.4)
Shared Bath Towels	12 (10.2)	8 (12.7)	4 (7.3)
Household Member Risk Factors			
Household Member with Recent SSTI	1 (0.8)	1 (1.6)	0 (0)
Household Member with Recent Surgery	7 (5.9)	4 (6.4)	3 (5.4)
Household Member Works in Healthcare	12 (10.1)	7 (11.1)	5 (8.9)

Note: No significant differences at baseline were observed between experimental and control groups at an alpha (2-tailed) = 0.05 with the exception of: (1) MRSA+ wound cultures, which were more common in the experimental group and MSSA+ wound cultures, which were more common in the usual care group (both ps = 0.0004), (2) USA 300 was more common in the experimental group (52%) as compared to the control group (32%, *p* = 0.03).

**Table 2 antibiotics-10-01105-t002:** SSTI recurrence in Index Patients at six-month follow-up ^a^.

	Total ^b^ (n = 119)	Experimental (n = 63)	Usual Care (n = 56)	OR (95% CI)	*p*-Value
Prospective, n (%):					
EHR-Based (n = 119)	13 (10.9) ^c^	7 (11.1)	6 (10.7)	1.14 (0.35, 3.6)	0.82
Self-Report (n = 92)	13 (15.3)	10 (22.2)	3 (7.5)	3.5 (0.89, 13.8)	0.07
Combined Measure (Either EHR-Based or Self-Report)	24 (20.3)	15 (24.2)	9 (16.1)	1.7 (0.66, 4.2)	0.27

Notes: ^a^ Prospective recurrence is defined as report of a new SSTI in the 6-month period following the initial (baseline) infection for which the participant was recruited. EHR-based outcomes were assessed at 6-months post-baseline and include the time period 12 months prior and 6 months after the baseline infection. At the baseline telephone assessment (T0), and prospective recurrence was assessed at the 6-month telephone assessment (T4). ^b^ Total n differs by data source: EHR-based (n = 119), Self-Report (n = 92). All analyses are unadjusted for covariates. ^c^ The observed prospective recurrence rate at 6-month EHR review for the Observation Only Group (n = 66, 10.5%) was not different from either the Experimental (11.1%) or Usual Care (10.7%) or Total (10.7%).

## Data Availability

Data will be made available on a case-by-case basis. Please contact the Corresponding Author for additional information.

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
