# Peer review of "Comparative Effectiveness Study of Home-Based Interventions to Prevent CA-MRSA Infection Recurrence"

_antibiotics, 2021, doi:10.3390/antibiotics10091105_

Round 1

Reviewer 1 Report

In their manuscript, Tobin et. al. present a study of homes-based interventions to help reduce MRSA spread and recurrence. Overall, the study design was robust based on the control and experimental intervention employed; however, the study itself seems underpowered to detect statistical significance. In addition, the authors need to further describe adherence and possible confounders from self0seeking or alternate medical advice as well as how generalizable the data is to the general population.

  • Please show statistical results )p value) for all factors indicated on tables for comparison
  • Table 1a, b, c should likely be 3 different Tables or only important factors should be shown in the main text and the rest moved to Supplementary material
  • Was colonization measured by only presence of bacterial pathogen (ie: Yes/No) or by actual titration of bacterial levels present in each swab? Bacterial load may provide a better idea of overall reductions in prevalence due to experimental parameters.
  • It is confusing that there was no differences in patient-centered outcomes between baseline and follow-up when educational materials were provided.
  • Overall, there were some obvious experimental confounders (as discussed by the authors in the limitation section). Is it possible to re-assess the data based on number of MRSA+ wounds to see if these future directions would provide statistically significant returns if individuals were randomized according to further testing before inclusion in experimental versus control group?
  • Please show the power calculation performed to assess the inclusion cristeria for the study and what this was powered on as the study itself seems underpowered for assessing the effect of intervention.
  • Was any measure taken for adherence to experimental protocols? What was the proportion of cases/households that were able to completely follow all intervention parameters versus those that were not? How many households under UC also performed other interventions based on alternate medical advice, self-seeking health behavior, etc…

Author Response

We are pleased to submit this revised Manuscript ID: Antibiotics-1349704 as a submission to the themed issue in honor of Professor Alexander Tomasz in recognition of his outstanding contributions to the fields of antibiotic resistance and bacterial infectious diseases.

We appreciate the thoughtful reviews and suggested revisions, and we have tried to address each one, with our responses summarized below, and as appropriate, in the revised manuscript. 

We hope these have been responsive to all of the suggestions and requested revisions.

Jonathan N. Tobin, PhD

President/CEO

Clinical Directors Network (CDN)

Co-Director, Community-Engaged Research

Senior Epidemiologist & Adjunct Professor

The Rockefeller University

On Behalf of the co-authors

REVIEWER 1

We thank Reviewer 1 for a positive overall review, and have addressed the recommendations as follows:

  1. …however, the study itself seems underpowered to detect statistical significance.

We thank the reviewer for this observation, and agree that we were certainly not powered to detect the observed differences (which were very small on the primary outcome).  However, as stated in the manuscript (starting in line 395), we did perform an a priori sample size calculation needed to detect what we considered to be both a clinically meaningful, and achievable effect, based upon prior research.  Unfortunately, we did not observe that level of change, and, appropriately, did not reject the null hypothesis of no difference.

  1. In addition, the authors need to further describe adherence and possible confounders from self0seeking or alternate medical advice

We agree that there that there may be additional confounders. Typically, unmeasured covariates would serve to decrease an observed effect; in this case, there was no effect to reduce. One possible reason that we did not observe a difference between conditions was the possibility that the usual care group sought care or attended to cleaning more.

  1. …as well as how generalizable the data is to the general population.

CAMP2 was conducted in large in a diverse urban settings that provide care to minority, immigrant and other underserved and low-income patients.  Generalizability is always a challenge with populations who volunteer for clinical trials (i.e., in this case, those at risk for MRSA/MSSA recurrence who consent to and complete a complex study such as CAMP2. Tables 1a and 1b provide an extensive description of the demographic and clinical characteristics of the study participants, and the results should be applied to similar populations.

  1. Please show statistical results (p value) for all factors indicated on tables for comparison

Starting in line 86 and on, we indicate in the footnotes to the tables that overall, we found no significant differences; we did highlight the few instances where there were baseline differences that were statistically significant.  We did not want to have readers focus on numerous statistical comparisons in the descriptive tables, and did not want to clutter the presentation of the large amount of information being presented with multiple non-significant p-values.

  1. Table 1a, b, c should likely be 3 different Tables or only important factors should be shown in the main text and the rest moved to Supplementary material

We agree with the reviewer’s recommendation, and we have shortened these tables and rearranged the data based on this suggestion.

  1. Was colonization measured by only presence of bacterial pathogen (ie: Yes/No) or by actual titration of bacterial levels present in each swab? Bacterial load may provide a better idea of overall reductions in prevalence due to experimental parameters.

We agree with Reviewer 1 that “titration of bacteria levels present in each swab may provide a better idea of overall reductions in prevalence due to experimental parameters” particularly when facing facultative pathogens such as S. aureus. This information would be of great interest, but unfortunately, in CAMP2, which utilized a commercial clinical lab for speciation, colonization was measured only by detection: presence vs absence of the strain.

  1. It is confusing that there was no differences in patient-centered outcomes between baseline and follow-up when educational materials were provided.

We agree with Reviewer 1 and were also surprised at the lack of differences in patient-centered outcomes and that we did not detect any impact of patient educational materials on patient-centered outcomes.

8. Overall, there were some obvious experimental confounders (as discussed by the authors in the limitation section). Is it possible to re-assess the data based on number of MRSA+ wounds to see if these future directions would provide statistically significant returns if individuals were randomized according to further testing before inclusion in experimental versus control group?

We agree with Reviewer 1 that examining number of MRSA+ wounds might identify subgroups with differential responses to the promotora-delivered intervention.  We report here on the pre-specified subgroups for the heterogeneity of treatment effects (HTE) as included in the initial analysis plans (see Lines 243-252).  Future manuscripts will examine additional factors related to intervention uptake, adherence and impact, such as SSTI burden as measured by the number of MRSA+ wounds. If there is evidence of effect of SSTI burden, future randomized clinical trials may wish to stratify prior to randomization on numbers of SSTI burden/number of MRSA+/MSSA+ wounds.

9. Please show the power calculation performed to assess the inclusion cristeria for the study and what this was powered on as the study itself seemed for assessing the effect of intervention.

In response to Reviewer #1’s request to show the power calculation, we direct the Reviewer to the sample size calculation/power analysis presented on page 17, lines 405-411, is copied here, and we added a sentence to clarify the both the initial and conditional power calculations, under the original (33%) and revised (11%) recurrence rates:

Sample size estimation was based on SSTI recurrence rate from CAMP1, where 33.3% experienced an recurrence during the six months following their index SSTI (21); previous studies of HA-MRSA reported reductions between 30% and 55% (48-50) To achieve 80% power at 5% significance level in a two-sided Chi Square/ Fisher’s Exact test for recurrence at 6-month follow-up, estimated sample size requirement was 120 participants [60 per group]. Revised sample size calculations based on a conditional power analysis using the lower observed event rate indicate that the study still had a power >0.80 to test a 50% reduction given the baseline recurrence rate of 11% (rather than 33%). [Added on page 17 lines 425 to 428].

10. Was any measure taken for adherence to experimental protocols? What was the proportion of cases/households that were able to completely follow all intervention parameters versus those that were not? How many households under UC also performed other interventions based on alternate medical advice, self-seeking health behavior, etc…

We agree with Reviewer 1 that valid measures of adherence to the experimental protocols would be useful to measure.  For example, we had initially considered assessing adherence to mupirocin use by weighing the tubes at the follow-up visit; however, our patient advisors and promotoras felt that obtaining this measure would “indicate a lack of trust” by the investigators towards the research participants, and for better or worse, we followed their recommendation.  We did attempt to assess adherence to antibiotics, topical mupirocin application, chlorhexidine bathing and household cleaning using self-reports; however, as we reported here that there was poor concordance between self-reported SSTI recurrence and SSTI recurrence as documented by clinicians in the EHRs (see lines 164 to 170).  For the sake of consistency in our analytic approach, since we suggested that the EHR data were more robust than self-report, we felt we could not make the reverse argument for the robustness of the self-reported adherence measures.

Reviewer 2 Report

The submitted manuscript is an interesting study to examine comparative effectiveness of usual care versus an experimental intervention (UC + universal decolonization + environmental decontamination). The study section and results are described in extensive details that will surely make reader capable to reproduce the study. Further the discussion correlates with the findings. The study will be interesting to the wide readers of Antibiotics specifically to the clinicians, epidemiologists and healthcare-associated scientists. My minor comments are:

  1. Elaborate the abbreviations on their first occurrences. Such as FQHCs.
  2. The authors have used several abbreviations in the manuscript. I recommend making a table of all the abbreviation to ease the reading.

  1. Figure 1: Please clean the figure. The current version is full of “?”

  1. Table 1a: “Length of time in the USA (if non-USA born) “

Is this duration continuous? Or Did the participant visit a foreign country in between?

How are the following parameter relevant to current study? RELIGIOUS AFFILIATION; LANGUAGE SPOKEN AT HOME; Frequency of Visits to Place of Worship

Author Response

We are pleased to submit this revised Manuscript ID: Antibiotics-1349704 as a submission to the themed issue in honor of Professor Alexander Tomasz in recognition of his outstanding contributions to the fields of antibiotic resistance and bacterial infectious diseases.

We appreciate the thoughtful reviews and suggested revisions, and we have tried to address each one, with our responses summarized below, and as appropriate, in the revised manuscript. 

We hope these have been responsive to all of the suggestions and requested revisions.

Jonathan N. Tobin, PhD

President/CEO

Clinical Directors Network (CDN)

Co-Director, Community-Engaged Research

Senior Epidemiologist & Adjunct Professor

The Rockefeller University

On Behalf of the co-authors

We thank Reviewer 2 for a positive overall review, and have addressed these recommendations as follows:

  1. Elaborate the abbreviations on their first occurrences. Such as FQHCs.

Thank you for this suggestion. We have revised the text in each location where an abbreviation is first used.

  1. The authors have used several abbreviations in the manuscript. I recommend making a table of all the abbreviation to ease the reading.

Thank you for this suggestion. We have added a table of abbreviations as Appendix H to improve clarity.

  1. Figure 1: Please clean the figure. The current version is full of “?”

We did not observe “?” symbols in Figure 1. In this revised version, the aberrant symbols have been fixed.

  1. Table 1a: “Length of time in the USA (if non-USA born) “ Is this duration continuous? Or Did the participant visit a foreign country in between?

We had published prior findings from the CAMP1 observational study (see Reference #41) that indicated a higher rate of MSSA among non-USA-born participants as compared to USA-born participants, so we pre-specified birthplace (USA-born, Non-USA-born) as a covariate to examine.  We agree that information about length of time residing outside the USA or visiting other countries after moving to the US would be interesting, but unfortunately, we did not capture more granular information on travel between NYC and other countries, or the duration of time spent in different locations, which might allow a more rigorous assessment of person-months of exposure.

  1. How are the following parameter relevant to current study? RELIGIOUS AFFILIATION; LANGUAGE SPOKEN AT HOME; Frequency of Visits to Place of Worship

These measures are part of a general demographic screening tool we have used in multiple prior studies. As we didn’t have any specific hypotheses related to religious affiliation or the frequency of visits to places of worship, we have removed these results from Table 1a.  We do capture language spoken at home as part of an assessment of acculturation, as well as to provide more refined description of the population recruited; so we have retained language spoken at home data in Table 1a.

Reviewer 3 Report

In the article “Comparative Effectiveness Study of Home-Based Interventions to Prevent CA-MRSA Infection Recurrence” Tobin et al have studied the community-based trial examined the effectiveness of an evidence-based intervention to reduce SSTI recurrence, mitigate household contamination/transmission, and improve patient-reported outcomes. The article is well presented and does have merits to be published in its original form. The article does not seem to have any flaws in collecting the information and data is presented in a scientific manner.

However, as a reader, I am a bit confused about the total outcomes or conclusions made by the authors. What is the takeaway message from this study? The message should be the part of abstract in the conclusion section.

From the trial study, what does the author suggests to the community or the caregivers?

Author Response

We are pleased to submit this revised Manuscript ID: Antibiotics-1349704 as a submission to the themed issue in honor of Professor Alexander Tomasz in recognition of his outstanding contributions to the fields of antibiotic resistance and bacterial infectious diseases.

We appreciate the thoughtful reviews and suggested revisions, and we have tried to address each one, with our responses summarized below, and as appropriate, in the revised manuscript. 

We hope these have been responsive to all of the suggestions and requested revisions.

Jonathan N. Tobin, PhD

President/CEO

Clinical Directors Network (CDN)

Co-Director, Community-Engaged Research

Senior Epidemiologist & Adjunct Professor

The Rockefeller University

On Behalf of the co-authors

We appreciate that Reviewer 3 describes the article as “well presented and does have merits to be published in its original form…. does not seem to have any flaws in collecting the information and data is presented in a scientific manner.”

  1. However, as a reader, I am a bit confused about the total outcomes or conclusions made by the authors. What is the takeaway message from this study? The message should be the part of abstract in the conclusion section.

The takeaway message is that our study confirms the effectiveness of the standard of care recommended by IDSA and CDC:  I&D with antibiogram-selected antibiotics.  In CAMP2, while there were small signals suggesting that the promotora-delivered intervention might be more effective, we cannot conclude this from the current study; other studies which used similar interventions delivered at a longer duration or repeated over time suggest that as similar intervention delivered at a higher dose than the one delivered in CAMP2 might be more effective, but future studies will need to examine this.

  1. From the trial study, what does the author suggests to the community or the caregivers?

We observed a similar recurrence rate between the two groups, suggesting that the primary guidelines-based approach of I&D and antibiotics may be sufficient to reduce the likelihood of recurrence.  Nevertheless, given that we observed high rates of environmental contamination with MRSA and MSSA in the households, efforts to minimize colonization and contamination, such as those provided by CAMP2, should be evaluated further. Future trials might examine and stratify randomization by low vs. medium vs. high colonization/contamination.

Reviewer 4 Report

This reviewer has no further comments on the overall research design, data collection and interpretation. Following are some minor comments for the authors' information to improve the quality of this manuscript.
1) The introduction section needs to be expanded extensively to show the background and the significance of this research.
2) Tables and figure needs further reorganization. 
3) Error code (?) in Fig. 1? 
4) For the discussion section, it might be better to be divided into several sub-section to clearly express the authors' logic and the significance of the results.

Author Response

n/a

Round 2

Reviewer 1 Report

The authors have taken the time to address my previous comments. Thank you.